# Determinants of Pulmonary Emphysema Severity in Taiwanese Patients with Chronic Obstructive Pulmonary Disease: An Integrated Epigenomic and Air Pollutant Analysis

**DOI:** 10.3390/biomedicines9121833

**Published:** 2021-12-04

**Authors:** Sheng-Ming Wu, Wei-Lun Sun, Kang-Yun Lee, Cheng-Wei Lin, Po-Hao Feng, Hsiao-Chi Chuang, Shu-Chuan Ho, Kuan-Yuan Chen, Tzu-Tao Chen, Wen-Te Liu, Chien-Hua Tseng, Oluwaseun Adebayo Bamodu

**Affiliations:** 1Division of Pulmonary Medicine, Department of Internal Medicine, Shuang Ho Hospital, Taipei Medical University, New Taipei City 235, Taiwan; chitosan@tmu.edu.tw (S.-M.W.); weilun.sun@tmu.edu.tw (W.-L.S.); leekangyun@tmu.edu.tw (K.-Y.L.); fengpohao@tmu.edu.tw (P.-H.F.); chuanghc@tmu.edu.tw (H.-C.C.); shu-chuan@tmu.edu.tw (S.-C.H.); 14388@s.tmu.edu.tw (K.-Y.C.); 09330@s.tmu.edu.tw (T.-T.C.); b7801077@tmu.edu.tw (W.-T.L.); 2Division of Pulmonary Medicine, Department of Internal Medicine, School of Medicine, College of Medicine, Taipei Medical University, Taipei 110, Taiwan; 3Graduate Institute of Clinical Medicine, College of Medicine, Taipei Medical University, Taipei 110, Taiwan; 4Department of Biochemistry and Molecular Cell Biology, Taipei Medical University, Taipei 110, Taiwan; cwlin@tmu.edu.tw; 5International PhD Program for Cell Therapy and Regeneration Medicine, Taipei Medical University, Taipei 110, Taiwan; 6TMU Research Center of Thoracic Medicine, Taipei Medical University, Taipei 110, Taiwan; 7School of Respiratory Therapy, College of Medicine, Taipei Medical University, Taipei 110, Taiwan; 8Institute of Epidemiology and Preventive Medicine, College of Public Health, National Taiwan University, Taipei 106, Taiwan; 9Department of Urology, Shuang Ho Hospital, Taipei Medical University, New Taipei City 235, Taiwan; 10Division of Hematology and Oncology, Department of Internal Medicine, Shuang Ho Hospital, Taipei Medical University, New Taipei City 235, Taiwan; 11Department of Medical Research, Shuang Ho Hospital, Taipei Medical University, New Taipei City 235, Taiwan

**Keywords:** chronic obstructive pulmonary disease, COPD, emphysema, severity, BMI, *lnc-IL7R*, PM_2.5_, PM_10_, SO_2_

## Abstract

Background: Chronic obstructive pulmonary disease (COPD) continues to pose a therapeutic challenge. This may be connected with its nosological heterogeneity, broad symptomatology spectrum, varying disease course, and therapy response. The last three decades has been characterized by increased understanding of the pathobiology of COPD, with associated advances in diagnostic and therapeutic modalities; however, the identification of pathognomonic biomarkers that determine disease severity, affect disease course, predict clinical outcome, and inform therapeutic strategy remains a work in progress. Objectives: Hypothesizing that a multi-variable model rather than single variable model may be more pathognomonic of COPD emphysema (COPD-E), the present study explored for disease-associated determinants of disease severity, and treatment success in Taiwanese patients with COPD-E. Methods: The present single-center, prospective, non-randomized study enrolled 125 patients with COPD and 43 healthy subjects between March 2015 and February 2021. Adopting a multimodal approach, including bioinformatics-aided analyses and geospatial modeling, we performed an integrated analysis of selected epigenetic, clinicopathological, geospatial, and air pollutant variables, coupled with correlative analyses of time-phased changes in pulmonary function indices and COPD-E severity. Results: Our COPD cohort consisted of 10 non-, 57 current-, and 58 ex-smokers (median age = 69 ± 7.76 years). Based on the percentages of low attenuation area below − 950 Hounsfield units (%LAA_-950insp_), 36 had mild or no emphysema (%LAA_-950insp_ < 6), 22 were moderate emphysema cases (6 ≤ %LAA_-950insp_ < 14), and 9 presented with severe emphysema (%LAA-950insp ≥ 14). We found that BMI, *lnc-IL7R*, PM_2.5_, PM_10_, and SO_2_ were differentially associated with disease severity, and are highly-specific predictors of COPD progression. Per geospatial levels, areas with high BMI and *lnc-IL7R* but low PM_2.5_, PM_10_, and SO_2_ were associated with fewer and ameliorated COPD cases, while high PM_2.5_, PM_10_, and SO_2_ but low BMI and *lnc-IL7R* characterized places with more COPD cases and indicated exacerbation. The prediction pentad effectively differentiates patients with mild/no COPD from moderate/severe COPD cases, (mean AUC = 0.714) and exhibited very high stratification precision (mean AUC = 0.939). Conclusion: Combined BMI, *lnc-IL7R*, PM_2.5_, PM_10_, and SO_2_ levels are optimal classifiers for accurate patient stratification and management triage for COPD in Taiwan. Low BMI, and *lnc-IL7R*, with concomitant high PM_2.5_, PM_10_, and SO_2_ levels is pathognomonic of exacerbated/aggravated COPD in Taiwan.

## 1. Introduction

Chronic Obstructive Pulmonary Disease (COPD), entailing small airways inflammatory disease and parenchyma destruction (otherwise known as emphysema), is a common, avoidable, and currently incurable respiratory pathology secondary to protracted and significant exposure to noxious gases and/or particulate matter [1,2]. The pathological hallmarks of emphysema include loss of lung tissue and accelerated loss of pulmonary function [3], where the latter is associated with reduced gas exchange, altered airway dynamics, impaired expiratory airflow, and progressive air trapping [4].

Protracted exposure to air pollutants, including oxides of nitrogen (NO_x_), ambient ozone (O_3_), emitted hydrocarbons (HC), and fine particulate matter < 2.5 μm in aerodynamic diameter (PM_2.5_), has been implicated in the progressive reduction of pulmonary function in patients with COPD/emphysema [5,6,7]. The significant increase in PM_2.5_ exposure-specific COPD burden observed over the last decade [8], cannot be decoupled from reported reduction in lung function indices, such as the forced vital capacity (FVC), forced expiratory volume in 1s (FEV1), maximum mid-expiratory flow (MMEF), and FEV1/FVC ratio, elicited by every 5 μg/m^3^ increase in PM_2·5_ [6]. 

While our understanding of the pathobiology of COPD has greatly increased in the last two decades, its prevalence, disease burden, and mortality remain unabated, pharmacological therapies continue to exhibit limited effects on morbidity and mortality, and it remains rather unclear why the presence and severity of emphysema differs significantly between patients with COPD [9].

Although tobacco smoking is widely reported to strongly influence predisposition to COPD, other environmental factors, including noxious gases and particulate matters, and endogenous (age, genetic, and epigenetic) factors, are increasingly documented as important determinants of COPD [10,11]. With only ~14% of the overall COPD burden attributable to occupational exposures [12], the relevance of endogenous factors in the pathogenesis of COPD is accentuated by reports indicating that globally, an estimated 25–45% of COPD cases were “never smokers” [12,13].

The last decade has been characterized by accruing evidence that tobacco smoking, which is considered a principal risk factor for COPD, alongside aging, is associated with epigenetic reprogramming of the bronchial epithelium and that epigenetic signal transduction pathways regulate COPD-related airway inflammation [14]. More so, Zeng H, et al. demonstrated that cigarette smoking elicits pulmonary cell death and B-cell lymphoma/leukemia-2 (Bcl-2) promoter hypermethylation in emphysema models through induction of oxidative stress and activation of epigenetic DNA methyltransferase enzyme 1 (DNMT1) [15]. Recently, our team reported that Toll-like receptor (TLR)-related long non-coding interleukin 7 receptor (*lnc-IL7R*) levels were significantly downregulated in the peripheral blood mononuclear cells of patients with COPD, compared with those from their healthy control peers.l This suppressed *lnc-IL7R* levels was also found to be associated with impaired pulmonary function, and increased risk of COPD exacerbation [16].

Against this background, the present study probed for pathognomonic biomarkers and/or clinicopathological parameters that determine the severity of emphysema in patients with COPD (COPD-E), affect disease course, and predict clinical outcome, while informing therapeutic strategy in Taiwan. Stemming from the working hypothesis that a multi-variable model rather than single variable model may be more pathognomonic of COPD-E, we demonstrate herein that a pentad comprising of body mass index (BMI), *lnc-IL7R*, ambient PM_2.5_, PM_10_, and SO_2_ concentrations were differentially associated with disease severity in Taiwanese patients with COPD-E, and are highly-specific predictors of COPD-E progression. For the first time, to the best of our knowledge, the present study demonstrated that per geospatial levels, areas with high BMI and *lnc-IL7R* but low ambient PM_2.5_, PM_10_, and SO_2_ were associated with fewer and ameliorated COPD-E cases, while high PM_2.5_, PM_10_, and SO_2_ but low BMI and *lnc-IL7R* characterized regions with more COPD-E cases and indicated disease exacerbation. The prediction pentad effectively differentiates patients with mild/no COPD from moderate/severe COPD cases, (mean AUC = 0.714) and exhibited very high stratification precision (mean AUC = 0.939).

## 2. Methods

### 2.1. Study Design and Patients

The study enrolled 168 subjects (patients with COPD, *n* = 125 and healthy participants, *n* = 43) who presented and underwent high resolution computed tomography (HRCT) to determine the presence and severity of COPD-E at the Department of Thoracic Medicine, Shuang Ho Hospital, Taipei Medical University (New Taipei, Taiwan) between March 2015 and February 2021. Image attenuation on the acquired HRCT scans of participants’ entire lung was assessed using the APOLLO workstation version 1.2 (VIDA Diagnostics Inc., Coralville, IA, USA) at a single reading center by two highly trained experts. Based on the percentages of low attenuation area below −950 Hounsfield units (%LAA_-950insp_), patients were classified as having no or mild emphysema (%LAA_-950insp_ < 6), moderate emphysema (6 ≤ %LAA-950insp < 14), and severe emphysema (%LAA_-950insp_ ≥ 14) [17]. COPD severity assessment was consistent with the Global Initiative for Chronic Obstructive Lung Disease (GOLD) guidelines [5].

The present single-center, prospective, two-arm, non-randomized study was approved by the Joint Institutional Review Board of Taipei Medical University (TMU-JIRB Approval No.: N201803059, N201902021), and was compliant with the Declaration of Helsinki guidelines on studies involving human subjects. Written informed consent was obtained from all participants before sample collection.

### 2.2. Inclusion and Exclusion Criteria

Inclusion criteria: Participants were included in the COPD arm if they were aged ≥ 40 years, had established COPD diagnosis, with a FEV1/FVC < 70% after administration of a bronchodilator, according to the GOLD criteria, in stable condition, with no COPD exacerbation in the last 1 month, had no diagnosis of cardiac disease, and accepted to participate by providing written informed consent. For the healthy arm, participants had FEV1 > 80% and FEV1/FVC > 75%, and no known systemic diseases in the previous 3 months.

Exclusion criteria: Participants were excluded if they had reversible airflow obstruction greater than 12% and 200 mL after inhalation of bronchodilator (according to the American thoracic Society (ATS) guidelines), had documented clinical history of previous or present asthma episodes, or coexisting abnormalities on the CT scan.

### 2.3. Geographic Information System (GIS) and Ambient Air Pollutant Exposure

Ambient air pollution data, namely, concentrations of PM_2.5_, PM_10_ (PM < 10 μm aerodynamic diameter), SO_2_, O_3_, NO_2_, NO, NO_X_, carbon monoxide (CO), total hydrocarbon content (THC), non-methane hydrocarbons (NMHC), and methane (CH_4_), were obtained from the Air Quality Monitoring (AQM) Networks of the Taiwan Environmental Protection Administration (https://airtw.epa.gov.tw/ENG/Sitemap.aspx; accessed 1 February 2021). PM concentration is uninterruptedly measured by all AQM stations and recorded hourly. PM exposure data per diem was assigned to participants based on their residential address. The mean daily concentrations of ambient PM_2.5_ and PM_10_ in the preceding years were computed for subsequent analyses. The nearest 3 AQM stations (stations I, II and III) were identified using the ArcGIS server software version 10.8.1 (ESRI, Redlands, CA, USA), and then air pollution data was extracted. The distance range between AQM stations I, II, III, and participants’ residential addresses was 0.36–8.73 (mean = 2.35), 1.28–14.33 (mean = 3.55), and 2.56–35.93 (4.73) km, respectively. The mean daily concentrations of PM_2.5_ and PM_10_ were determined using the inverse distance weighting (IDW) interpolation method.

### 2.4. Sample Preparation and Quantitative Reverse Transcription PCR (RT-qPCR)

After the isolation of peripheral blood mononuclear cells (PBMC) from the whole blood of patients with COPD and healthy participants strictly following the Ficoll—Hypaque density gradient centrifugation protocol previously described by Chen TT, et al. [18], total RNA was isolated from the cell samples using the TRIzol™ Plus RNA Purification Kit (Cat. #12183555; Thermo Fisher Scientific, Waltham, MA, USA). Serum RNA was purified using the Plasma/Serum Circulating and Exosomal RNA Purification Kit (Slurry Format) (Cat. #42800; Norgen Biotek Corp., Thorold, ON, Canada). Following determination of total RNA concentration using the NanoDrop ND1000 spectrophotometer (Nyxor Biotech, Paris, France), PCR mix were prepared using the SYBR™ Green PCR Master Mix (Cat. #4309155; Applied Biosystems Inc., Carlsbad, CA, USA). The PCR contained the primers, the fluorogenic probe mix, and the TaqMan Universal PCR Master mix (Applied Biosystems Inc.). All amplification reactions were performed in quadruplicates from 20 ng complementary DNA (cDNA) in the Bio-Rad C1000 real-time PCR system (Bio-Rad, Cambridge, MA, USA) using the following conditions: 95 °C for 3 min, 35 cycles at 95 °C for 15 s, 60 °C for 30 s, 72 °C for 30 s, and 72 °C for 10 min. For analysis of results, all values were normalized to the levels of the housekeeping genes *18S rRNA* (cellular) or *GAPDH* (extracellular), which served as the internal control. All procedures were consistent with manufacturers’ instructions, and the following PCR primer sequences were used: *lnc-IL7R* (forward): 5′-CCAGCCTTTGCCTCTTCCTTCAAT-3′, *lnc-IL7R* (reverse): 5′-CCGTA CCAAGTCTCTTAGCCCCTC-3′; *18S* rRNA (forward): 5′-TGTGCCGCTAGAGGTGAAATT-3′, *18S* rRNA (reverse): 5′-TGGCAAATGCTTTCGCTTT-3′; *GAPDH* (forward): 5′-ATGGGGAAGGTGAAGGTCG-3′, and *GAPDH* (reverse): 5′-GGGGTCATTGATGGCAACAAT-3′.

### 2.5. Statistical Analysis

Results are expressed as means ± standard deviations. Pearson’s chi-squared (χ^2^) test was used to determine the relationship or association between categorical variables. The paired *t*-test was used for comparing continuous data. The Student’s *t*-test was used to assess alterations in pulmonary function based on %LAA_-950insp_, concentration levels of ambient air pollutants, and *lnc-IL7R* expression levels. *p*-value ≤ 0.05 defined statistical significance. All statistical analyses were performed using IBM SPSS Statistics for Windows, Version 25.0 (IBM Corp. Released 2017, Armonk, NY, USA: IBM Corp). Geospatial visualization and analysis were performed using the ArcGIS server software version 10.8.1 (ESRI, Redlands, CA, USA).

## 3. Results

Emphysema (COPD-E) severity correlates with GOLD stage, and is indicative of disease progression in Taiwanese patients with COPD.

Table 1 contains the baseline characteristics of our cohort of patients with COPD (*n* = 125) and healthy controls (*n* = 43). As shown in Table 1, the median age was 69 ± 7.76 years (range: 41–87), 88.8% of all patients with COPD were male and aged 41–87 years. Of these, 6.31% were never-smokers, while 47.75% and 45.94% were ex- and current smokers, respectively. The median BMI was of 20.0 ± 4.46 kg/m^2^, and 23.66 ± 3.99 kg/m^2^ for female and male COPD cases. Stratification of patients with COPD according to the GOLD-based staging showed that compared with GOLD stage I cases with median FEV_1_ of 84.55 ± 7.13%, there was 23.1-fold (median FEV_1_ = 65 ± 9.27%), 53.8-fold (median FEV_1_ = 39.05 ± 5.53%), and 70.4-fold (median FEV_1_ = 25 ± 3.85%) decrease in the pulmonary function indices of their GOLD stages II, III, and IV peers, respectively. Regardless of smoking status, compared with the healthy control (median FEV_1_: non-smoker = 101.00 ± 5.10 vs smoker = 96.85 ± 6.68), significant decline in lung function was observed in the non-stratified COPD group (median FEV_1_ = 57 ± 19.16%) (Table 1).

Furthermore, based on the working hypothesis that a phenotype-based categorization of COPD which takes into account presence/severity of COPD-E and exacerbation history, rather than solely on level of airflow limitation (FEV_1_) better reflects the nosological complexity and constitutive heterogeneity of COPD, based on %LAA_-950insp_, we found 66.67% and 33.33% of the GOLD 1 cases had mild and moderate COPD-E, respectively. 72.73% of all patients with severe COPD-E were GOLD 3–4, 27.27% were GOLD 2, and 0.00% were GOLD 1 (Table 1). Conversely, all mild COPD-E cases were GOLD 2 (100%), while 17.24% of all moderate COPD-E was found to be GOLD 1 (Table 1). These data do indicate that COPD-E is not only diagnostic of COPD, but is also indicative of disease progression or exacerbation.

### 3.1. Emphysema Risk Modulators in Taiwanese Patients with COPD

Observing that COPD-E reflects the nosological complexity and disease progression in patients with COPD, we explored for COPD-E-specific determinants of disease severity, and treatment success in Taiwanese patients with COPD. Establishing a significantly strong correlation between the imaging emphysema severity indices, total LAA% and %LAA_-950insp_ (r = 0.82, *p* < 0.001) (Figure 1A), we next probed for probable correlation between emphysema severity and ambient air pollutants, epigenetic, spirometric, anthropometric, and clinical variables. The %LAA_-950insp_ was inversely correlated with epigenetic *lnc-IL7R* (r = −0.30, *p* = 0.002), post-bronchodilator FEV_1_/FVC (r = −0.41, *p* < 0.001), and BMI (r = −0.47, *p* < 0.001) (Figure 1B–D). Ambient PM_2.5_ (r = 0.31, *p* < 0.001), PM_10_ (r = 0.30, *p* = 0.001), NO_2_ (r = 0.18, *p* = 0.043), SO_2_ (r = 0.25, *p* = 0.004), and THC (r = 0.20, *p* = 0.024) concentrations were positively correlated with %LAA_-950insp,_ while O_3_ (r = −0.09, *p* = 0.341) was inversely correlated (Figure 1E–K). In addition, and equivocal association was found between %LAA_-950insp_ and pack-year (r = 0.17, *p* = 0.057) or age (r = 0.11, *p* = 0.240) (Figure 1L,M). These data are suggestive of a multifactorial risk modulatory cluster for COPD-E in Taiwanese patients with COPD.

### 3.2. Delineating Predictors of Disease Severity in Taiwanese Patients with COPD-E

Consistent with earlier data, using a supervised machine learning algorithm, artificial neural network (ANN) modelling based on the hyperbolic tangent activation (TanH) model with random holdback validation showed that ambient air pollutants (PM_2.5_, PM_10_, NO_2_, SO_2_, THC, O_3_), epigenetic (*lnc-IL7R*), anthropometric (Age, BMI), lifestyle (smoking history, pack-year), and geospatial components (longitude and latitude of participants’ residential addresses) all contribute differentially to emphysema (COPD-E) severity (%LAA_-950insp_) (Figure 2A,B). The initial fitting of the ANN model on training dataset (80% of our COPD cohort) for variable selection and parameter estimation to predict %LAA_-950insp_ indicated strong association between listed variables and COPD-E severity (R^2^ = 0.49; root mean square error, RMSE = 0.64; mean absolute deviation, MAD = 0.460) (Figure 2A). This was validated by the unbiased evaluation and hyper-parameter fine-tuning of the training dataset-fitted ANN model using the validation dataset (20% of our COPD cohort) (R^2^ = 0.65; root mean square error, RMSE = 0.60; MAD = 0.37) (Figure 2A). Interestingly, our prediction profile analysis indicate that BMI < 23.62 kgm^−2^, blood *lnc-IL7R* level < 0.53, smoking history (ex or current) with pack-year > 50.58, SO_2_ > 3.08 parts per billion (ppb), O_3_ < 26.27 ppb, NO_2_ > 20.08 ppb, THC < 2.18 parts per million (ppm), PM_10_ < 40.10 mg m^−3^, and PM_2.5_ > 22.18 mg m^−3^ define an exacerbated COPD-E phenotype (Figure 2B). Though not optimal, goodness of fit analysis showed that the ANN model can relatively predict COPD-E (%LAA_-950insp_) severity fairly accurately (RMSE = 0.58, R^2^ = 0.46, *p* < 0.0001) (Figure 2C). In parallel unsupervised machine learning analysis using the hierarchical clustering model, the COPD cohort was pooled into three clusters reflecting COPD-E severity, with cluster 0, 1, and 2 representing no emphysema, mild-moderate emphysema, and moderate-severe emphysema (R^2^ = 0.39; Pearson’s g = 0.47; Dunn index = 0.17; Silhoutte = 0.250; Entropy = 0.41, Calinski-Harabasz index = 20.84) (Figure 2D). The relatively low R^2^, Dunn index, Pearson’s g, and silhouette index, coupled with high entropy score necessitated model optimization. For optimal variable selection, the effect of BMI (t-ratio = −5.75, F-ratio = 33.01, *p* < 0.0001), *lnc-IL7R* (t-ratio = −2.07, F-ratio = 4.28, *p* = 0.042), and SO_2_ (t-ratio = 2.00, F-ratio = 3.99, *p* = 0.049) on COPD-E severity were statistically significant, while smoking history (t-ratio = 1.94, F-ratio = 3.75, *p* = 0.056), PM_10_ (t-ratio = −1.51, F-ratio = 2.30, *p* = 0.133), and PM_2.5_ though exhibiting good effect (F-ratio ≥ 2), were statistically non-significant (Figure 2E,F).

### 3.3. Severity-Stratified Spatiofunctional Interaction between Individual Predictors of COPD-E

To determine if and to what extent these COPD-E-associated factors interact and/or form a pathogenic cascade, we performed a bootstrapped network analysis (bootstrap *n* = 1000). Of pathophysiological relevance, we found no interaction between any of the COPD-E pathognomonic factors in patients with no or mild emphysema (%LAA_-950insp_ < 6) (Figure 3A, left). For the moderate emphysema cases (6 ≤ %LAA_-950insp_ < 14), we found that ambient SO_2_, NO_2_, THC, PM_10_, PM_2.5_, and O_3_ form a loose cascade with endogenous factors BMI, *lnc-IL7R*, and Age in the context of patients’ geolocation, while smoking history and pack-year were non-contributors to the cascade (Figure 3A, middle). Conversely, a tight-knit cluster was observed between all variables in patients with severe COPD-E (%LAA-950insp ≥ 14), with relatively high interaction density (Figure 3A, right). This partial deciphering of the structural and functional networks of COPD-E on a spatiotemporal scale, was corroborated by the Barrat, Onnela, Watts and Strogatz (WS), and Zhang clustering coefficients [19,20], which measure the propensity to which the nodes/factors tend to cluster together, and quantify the abundance of connected triangles in the weighted severity-based pathogenic networks (Figure 3B). Next, exploiting the ability of the centrality plot to identify important nodes/variables that determine disease severity, as well as the nature of their influence, we identified SO_2_, PM_10_, PM_2.5_, O_3_, and *lnc-IL7R tetrad* as an important predictor of disease progression and severity, and that while SO_2_, PM_10_, and PM_2.5_, drove disease progression and were positively correlated with COPD-E severity, O_3_ and *lnc-IL7R* attenuated progression and were negatively associated with disease severity (Figure 3C).

### 3.4. BMI, lnc-IL7R, PM_2.5_, PM_10_, and SO_2_ Levels Are Excellent Classifiers for Accurate Patient Stratification and COPD-E Management Triage in Taiwan

Finally, receiver operating characteristic (ROC) curve analysis was used to evaluate the discriminatory power of all COPD-E-associated factors, namely geospatial variables (longitude, latitude), ambient particulate matter and pollutants (PM_2.5_, PM_10_, SO_2_, O_3_, THC, NO_2_), lifestyle (pack-year), anthropometric (BMI), and endogenous (age, epigenetic *lnc-IL7R*) factors. All factors, except latitude (area under the ROC curve, AUC = 0.55), age (AUC = 0.51), O_3_ (AUC = 0.58), and pack-year (AUC = 0.57), exhibited acceptable capability to discriminate between patients with COPD-E and those with non-emphysematous COPD phenotype, regardless of severity status (Figure 4A,B). In parallel analysis of our COPD cohort using precision–recall curves, we found the latitude (AUC = 0.89, associated criterion ≤ 25.06), longitude (AUC = 0.93, associated criterion ≤ 121.80), SO_2_ (AUC = 0.95, associated criterion ≥ 2.14), THC (AUC = 0.94, associated criterion ≥ 2.14), PM_2.5_ (AUC = 0.95, associated criterion ≥ 16.24), PM_10_ (AUC = 0.95, associated criterion ≥ 16.24), BMI (AUC = 0.95, associated criterion ≤ 29.36), and *lnc-IL7R* (AUC = 0.91, associated criterion ≤ 1.23) excellently stratified patients into COPD-E and non-COPD-E groups (Figure 4C–G). Since a high AUC represents both high recall (related to low false negative rate) and high precision (associated with low false positive rate), where 0.5 denotes a bad classifier and 1, an excellent classifier, these results indicate that BMI, *lnc-IL7R*, PM_2.5_, PM_10_, and SO_2_ levels are excellent classifiers for accurate patient stratification and management triage for COPD-E in Taiwan.

### 3.5. BMI, lnc-IL7R, PM_2.5_, PM_10_, and SO_2_ Are Highly Specific Predictors of COPD-E Severity and Disease Progression in New Taipei City

Furthermore, predictor screening and effect ranking for predictive factor optimization showed that the highest ranked contributors to or determinants of emphysema (COPD-E) severity were BMI (contribution: 2.31, portion: 0.21, rank: 1), *lnc-IL7R* (contribution: 2.04, portion: 0.19, rank: 2), PM_2.5_ (contribution: 1.33, portion: 0.12, rank: 3), PM_10_ (contribution: 0.98, portion: 0.09, rank: 4), and SO_2_ (contribution: 0.81, portion: 0.07, rank: 5) (Figure 5A). Next, using the Gaussian Processes (GP) generic supervised learning algorithm which is designed to solve regression and probabilistic classification problems, we generated contour and surface profile plots for the GP prediction model of COPD-E (%LAA_-950insp_) severity. First, for geospatial relevance, we found that most non-COPD-E or mild/ameliorated COPD-E cases, as defined by %LAA_-950insp_, were resident above longitude 121.496° along latitude 24.996°, while the largest proportion of exacerbated or severe COPD-E cases were resident beyond this geographical level (Figure 5B). These exacerbated or severe COPD-E cases were defined by serum *lnc-IL7R* levels ≤ 0.54, BMI ≥ 23.54 kgm^−2^, and resident under ambient PM_2.5_ ≥ 22.48 mg/m^3^, PM_10_ ≥ 40.46 mg/m^3^, and SO_2_ ≥ 3.13 ppb (Figure 5C,D). More so, understanding that sensitivity implies the ability of any test to designate a subject with COPD-E as positive, our GP model of COPD-E severity showed that PM_2.5_ (total sensitivity = 0.833, theta = 1.70 × 10^−9^, nugget = 0.001), longitude (total sensitivity = 0.997, theta = 0.005, nugget = 0.001), BMI (total sensitivity = 0.633, theta = 8.21 × 10^−6^, nugget = 0.001), and SO_2_ (total sensitivity = 0.317, theta = 8.75 × 10^−5^, nugget = 0.001) are independent determinants or drivers of COPD-E in New Taipei City, Taiwan (Table 2, Table 3 and Table 4). Since the lower the sensitivity, the greater the ability of the GP model to designate a case without COPD-E as negative, we confirmed that *lnc-IL7R* (total sensitivity = 0.167, theta = 0.000, nugget = 0.001), and latitude (total sensitivity = 0.001, theta = 0.000) are factors of amelioration in the specified region (Table 2, Table 3 and Table 4). The very low theta values and nugget parameter are of statistical and predictive relevance, as they rule out measurement error and short scale variability [21].

### 3.6. Low BMI, and lnc-IL7R, with Concomitant High PM_2.5_, and SO_2_ Levels Is Pathognomonic of Exacerbated/Severe COPD-E in New Taipei City, Taiwan

For geospatial contextualization and visualization of the effect of the delineated determinants of COPD-E severity in New Taipei City, after digitizing our study areas at landscape, regional, and national scales on a global map as polygons, site and cases were marked as point features. The generated site-of-interest map is shown in Figure 6A, with COPD-E cases were concentrated around the Wanhua, Banqiao, Tucheng, Xinzhuang, Hsin-tien, Zhonghe, and Yonghe Districts. Consistent with Figure 5, the most severe COPD-E cases were in the Wanhua, Banqiao, Zhonghe, and Yonghe Districts, located below longitude 121.496° latitude 24.996°, with %LAA_-950insp_ ≥ 10 (i.e., ≥ 1.3 on a statistical scale of 1–3) (Figure 6B), ambient PM_2.5_ > 24.0 mg/m^3^ (Figure 6C), and SO_2_ > 3.5 ppb (Figure 6D), coupled with endogenous *lnc-IL7R* levels < 0.8 (Figure 6E), and BMI < 25.0 kgm^−2^ (Figure 6F). These data indicated that low BMI, and lnc-IL7R, with concomitant high PM_2.5_, and SO_2_ levels is pathognomonic of exacerbated/severe COPD-E in New Taipei City, Taiwan.

## 4. Discussion

Severe emphysema (COPD-E) remains a therapeutic challenge, especially in the light of the limited efficacy of contemporary anti-COPD therapeutic strategy. The present study accentuates the role of a multifactorial risk modulatory cluster for development and/or progression of COPD-E in Taiwanese patients with COPD. We showed that ambient air pollutants (PM_2.5_, PM_10_, NO_2_, SO_2_, THC, O_3_), epigenetic (*lnc-IL7R*), anthropometric (Age, BMI), lifestyle (smoking history, pack-year), and geospatial components (longitude and latitude of participants’ residential addresses) all contribute differentially to emphysema (COPD-E) severity (%LAA_-950insp_). This is in part consistent with reports by Wang M, et al. suggesting a significant association between observed increase in COPD-E severity over time and baseline concentration of ambient PM_2.5_, NO_X_, THC, or O_3_ [7].

While we cannot fully explain the inverse correlation between COPD-E status/severity and ambient O_3_ concentration in our study, we posit that this may be corollary to the significantly enhanced concentration of PM_2.5_ in the study sites, and this rationalization would be consistent with results of a recent study showing that a 40% reduction in PM_2.5_ over a period of 5 years in the North China Plains was in part responsible for a 1–3 ppb annual increase in O_3_ observed in megacity clusters of eastern China [22]. It is also possible that the geospatial localization of the study sites allows for enhanced natural influx of “good O_3_′ from the stratosphere into the troposphere, due to heightened vertical air movements, causing this “good O_3_′ to contribute immensely to the background concentration of ground-level O_3_ in the districts of New Taipei City specified in the present study [23]. Thus, rather than the increased ambient O_3_ concentrations exerting detrimental effects such as “breathing problems, triggering asthma attacks, reducing lung function, and increasing incidence of respiratory diseases” [23], which are characteristic of troposphere-O_3_-associated COPD-E, an inversely correlation was found between the O_3_ and COPD-E, suggesting a protective effect consistent with those reported by Alberto Hernández’s team in the context of COVID-19 [24].

The BMI, indirectly representing an individual’s degree of obesity, is a vital indicator or determinant of the phenotypic expression of COPD, its course, and prognosis [25,26,27,28]. Our study found that high BMI was associated with less incidence and severity of COPD-E. While this contradicts the prevalent forgone conclusion that obesity or high BMI is associated with disease exacerbation or progression in an array of theme-relevant publications [25,26,27,28], our finding is consistent with recent reports that patients with COPD with high BMI exhibit reduced dyspnea symptoms, relatively better lung function, and quality of life, health-wise [28]. More so, our findings corroborate those from a nationwide analysis of the Taiwan Obstructive Lung Disease study data from 12 hospitals in Taiwan, which showed that high BMI (BMI ≥ 24 kgm^−2^) is associated with a lower frequency of COPD exacerbation in Taiwan [29]. Furthermore, a systematic review of available literature supporting the evolving obesity paradox in COPD, concluded that compared with normal BMI (18.5–24.9 kgm^−2^), “low BMI is a risk factor for accelerated lung function decline, whilst high BMI has a protective effect” [30].

The present study provides some evidence that BMI, *lnc-IL7R*, PM_2.5_, PM_10_, and SO_2_ levels are excellent classifiers for accurate patient stratification and management triage for COPD-E in Taiwan. Our results revealed that emphysema in patients with COPD is positively correlated with particulate matter and noxious gases exposure, and that high-level exposure to PM_2.5_, PM_10_, and SO_2_ causes greater decline in pulmonary function. Interestingly, we also showed that alongside the ambient pollutants, endogenous *lnc-IL7R* is a highly specific predictors of COPD-E severity and disease progression in New Taipei City. This is corollary to our previously published work indicating that downregulated expression of plasma or tissue *lnc-IL7R* in patients with COPD enhances inflammation and is associated with acute exacerbation, and more so frequently [16]. Moreover, *lnc-IL7R* RNA expression in the serum and lung tissues of patients with COPD-E was positively correlated with BMI, but negatively correlated with PM_2.5_, PM_10_, and SO_2_ exposure. This aligns with the assumption that the large variability in COPD-E onset and progression is driven principally by a compound gene–environment cascade. As succinctly put by Devadoss et al., “the transcriptomic and epigenetic memory potential of lung epithelial and innate immune cells drive responses, such as mucus hyperreactivity and airway remodeling, that are tightly regulated by various molecular mechanisms, for which several candidate susceptibility genes have been described” [31]. We posit that by interacting with and suppressing *lnc-IL7R* expression, the ambient PM_2.5_, PM_10_, SO_2_, facilitates aryl hydrocarbon receptor (AHR)-mediated CYP1A1 activation, enhances generation of reactive oxygen species (ROS), induces oxidative stress with associated inflammatory responses, and consequently elicits chronic inflammatory diseases, including COPD-E [32]. This is corroborated by our data suggesting that *lnc-IL7R* induction may have a potential regulatory role in normal bronchial cells exposed to PM_2.5_ because endogenous *lnc-IL7R* expression was upregulated in normal but not COPD lung epithelial cells or PBMC, and lower *lnc-IL7R* expression was associated with emphysema in COPD (unpublished data).

Furthermore, our finding indicating that Low BMI, and *lnc-IL7R*, with concomitant high PM_2.5_, and SO_2_ levels is pathognomonic of exacerbated/severe COPD-E in New Taipei City, Taiwan, corroborates the conclusions of a recent comprehensive analysis of two large cohorts that a panel of disease-related variables improve predictive value for disease outcomes, compared with lone clinical variables and individual biomarkers [33].

Against the background of the findings documented herein, we are cognizant of and do point out that there are suggestions that chronic or longterm exposure, rather than momentary exposure to ambient air pollution significantly affects the incidence and prevalence of both emphysematous and non-emphysematous COPD, however, such reports remain inconclusive, howbeit with plausible biological mechanisms [34]. Consistent with conclusions drawn by Tamara Schikowski and her team [34], there is probable substantial evidence for a causal link between ambient air pollution and development of COPD, to the extent that diminished pulmonary function earlier in life translates into COPD with age. While we concur that a case for causality may be substantiated should the critical role of ambient air pollutants in repeated or cumulative exacerbation be considered in the development of COPD-E [3,5,10,34,35], the extent to which such cumulative or long-term exposure to ambient air pollution outweighs current or short-term exposure, in the development of COPD-E remains largely inconclusive.

### Limitations

As with studies of this nature, the present study has some limitations. First, the use of a relatively small sample size (*n* = 168) and single-center nature of the study may harbor suggestions of high variability and low reliability of the findings reported herein. A larger cohort from a multi-center setting with varied characteristics may be warranted for more accurate representation of the disease population, and for deriving robustly generalizable inferences. This will help to establish more accurate prediction tools and clinical decision support systems for COPD-E management. Secondly, considering the heterogeneous nature of the classifiers, namely ambient air pollutants (PM_2.5_, PM_10_, NO_2_, SO_2_, THC, O_3_), epigenetic (*lnc-IL7R*), anthropometric (Age, BMI), lifestyle (smoking history, pack-year), and geospatial components (longitude and latitude of participants’ residential addresses), the inclusion of a bridging factor such as blood oxidative stress marker may have been appropriate for mechanistic insight; however, this was omitted. Studies on this bridging factor is currently ongoing.

## 5. Conclusions

In conclusion, combined BMI, *lnc-IL7R*, PM_2.5_, PM_10_, and SO_2_ levels are optimal classifiers for accurate patient stratification and management triage for COPD-E in Taiwan. Low BMI, and *lnc-IL7R*, with concomitant high PM_2.5_, PM_10_, and SO_2_ levels is pathognomonic of exacerbated/aggravated COPD-E in Taiwan. These findings may help inform management efforts and environmental health policy formulation for lowering disease risk and severity in Taiwan.

## Figures and Tables

**Figure 1 biomedicines-09-01833-f001:**
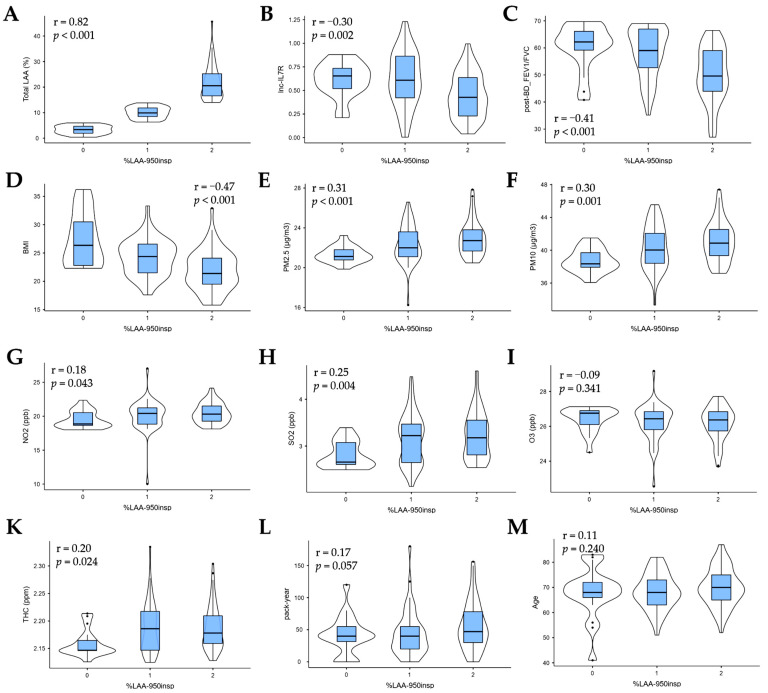
Disease risk modulators in Taiwanese patients with COPD-E. Violin plots with included box plots showing the correlation between (**A**) Total LAA, (**B**) *lnc-IL7R*, (**C**) post-bronchodilator FEV1/FVC ratio, (**D**) BMI, (**E**) PM_2.5_, (**F**) PM_10_, (**G**) NO_2_, (**H**) SO_2_, (**I**) O_3_, (**K**) THC, (**L**) pack-year, or (**M**) age, with %LAA_-950insp_-based COPD-E severity. COPD-E, emphysema in patients with COPD; r, Pearson correlation coefficient; %LAA_-950insp_, percentage of lung attenuation area with values less than− 950 Hounsfield units on inspiratory CT scan. For %LAA_-950insp_: 1, no or mild emphysema (%LAA_-950insp_ < 6); 2, moderate emphysema (6 ≤ %LAA_-950insp_ < 14); and 3, severe emphysema (%LAA_-950insp_ ≥ 14).

**Figure 2 biomedicines-09-01833-f002:**
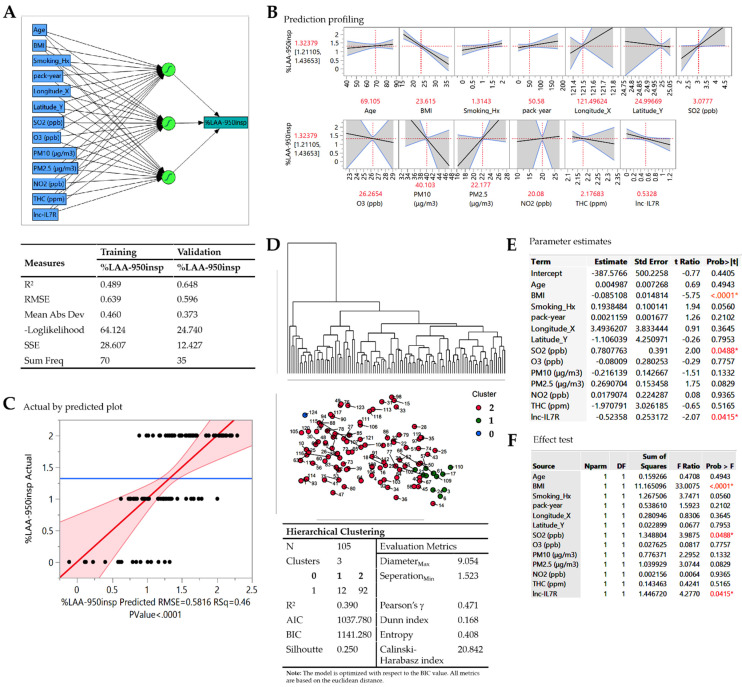
Delineating predictors of disease severity in Taiwanese patients with COPD-E. (**A**) Artificial neural network (ANN) model schema (upper panel) and statistics chart (lower panel) showing the 13 disease-related variables of interest input, 3 auto-determined hidden, and predicted %LAA_-950insp_-based COPD-E severity output layers. (**B**) Prediction profiler showing the effect of age, BMI, smoking history, pack-year, longitude, latitude, SO_2_, O_3_, PM_2.5_, PM_10_, NO_2_, THC, and *lnc-IL7R* on predicted %LAA_-950insp_-based COPD-E severity. Cut-off values are indicated in red. (**C**) Actual vs. predicted %LAA_-950insp_-based COPD-E severity plot. (**D**) Hierarchical clustering dendrogram and statistics of severity-stratified COPD-E cases. (**E**) Parameter estimates, and (**F**) effect test charts of the panel of variables. R^2^, coefficient of determination; RMSE, root mean square error; SSE, sum of squared estimate of errors; Hx, history; ppb, part per billion; ppm, part per million; N, number of cases; AIC, Akaike’s Information Criteria; BIC, Bayesian Information Criteria; Values in red, statistically significant.

**Figure 3 biomedicines-09-01833-f003:**
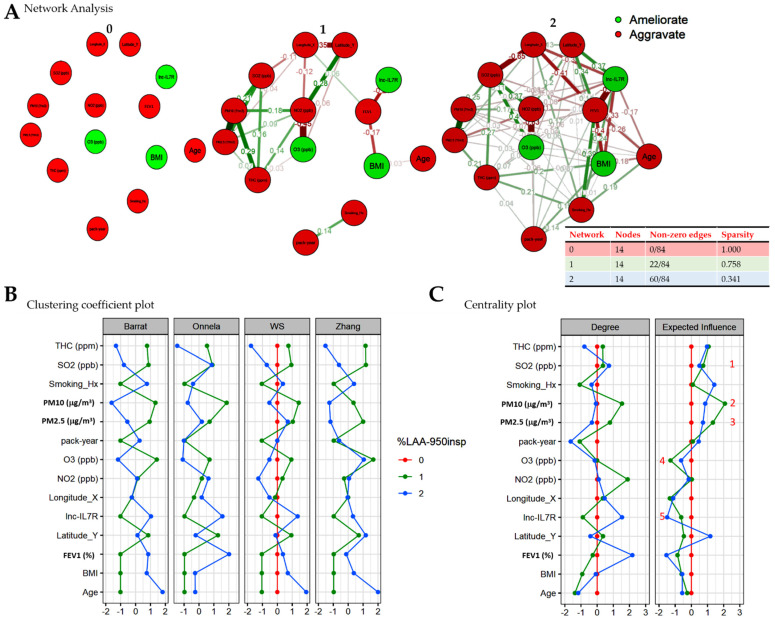
Severity-stratified spatiofunctional interaction between individual predictors of COPD-E progression in Taiwan. (**A**) Visualization, (**B**) clustering coefficient plot, and (**C**) centrality plot of the interaction between indicated variables in the determination of COPD-E severity as shown by bootstrapped network analysis. Number of bootstraps = 1000. red circle, aggravating factor; green circle, ameliorating factor. WS, Watts and Strogatz.

**Figure 4 biomedicines-09-01833-f004:**
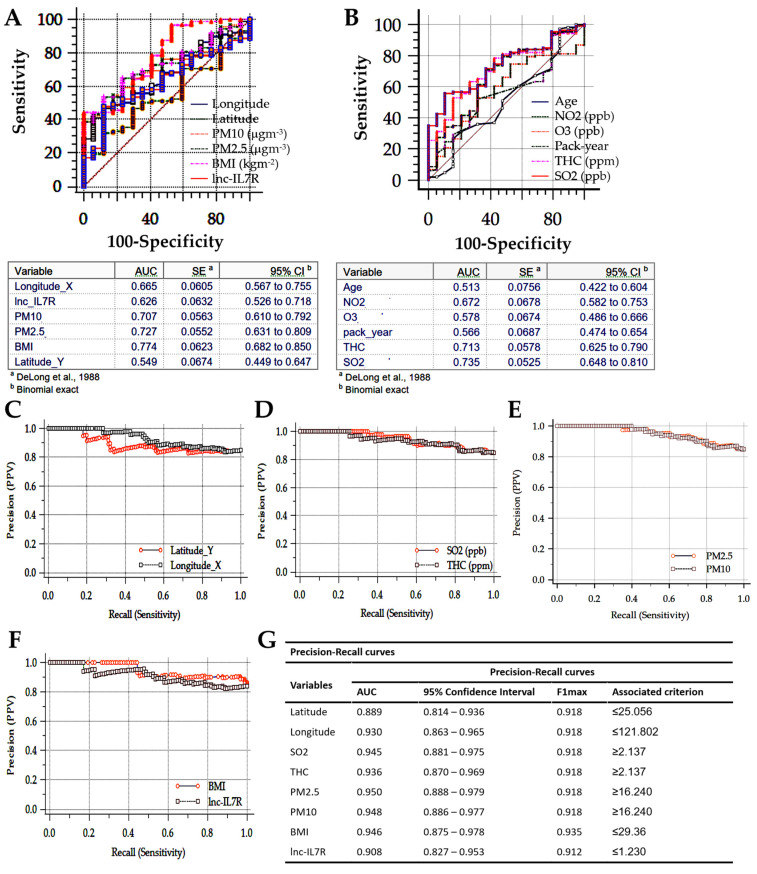
Combined BMI, *lnc-IL7R*, PM_2.5_, PM_10_, and SO_2_ levels are optimal classifiers for accurate patient stratification and COPD-E management triage in Taiwan. Visual representation (upper panel) and statistics chart (lower panel) of the area under the receiver operating characteristics curves for (**A**) longitude, latitude, PM_10_, PM_2.5_, BMI, *lnc-IL7R*, (**B**) age, NO_2_, O_3_, pack-year, THC, and SO_2_. Precision–recall curves showing the tradeoff between precision and recall for (**C**) longitude, latitude, (**D**) THC, SO_2_, (**E**) PM_2.5_, PM_10_, (**F**) BMI, and *lnc-IL7R*. (**G**) Statistics chart of the precision–recall curves for (**C**–**F**).

**Figure 5 biomedicines-09-01833-f005:**
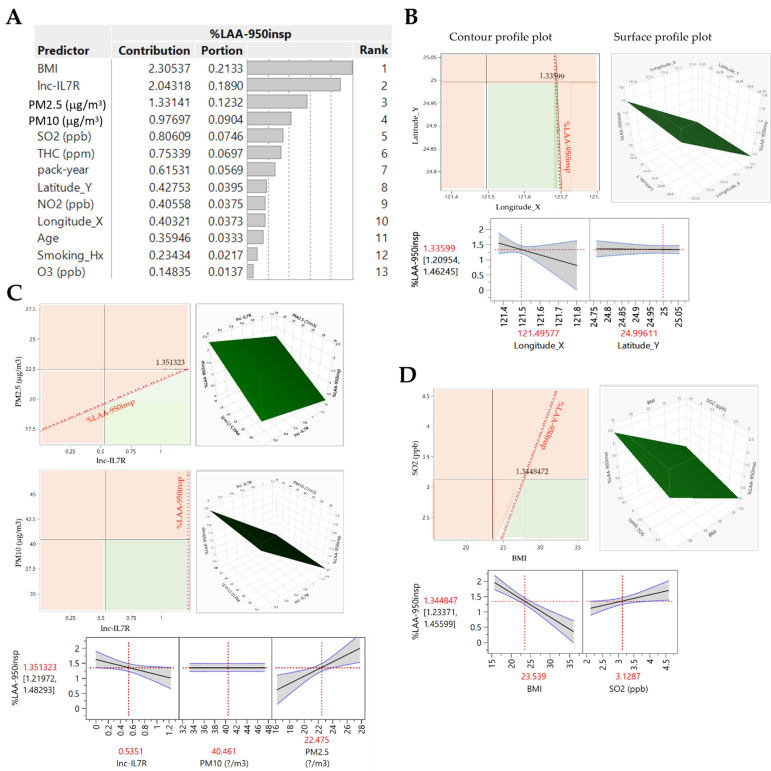
BMI, lnc-IL7R, PM_2.5_, PM_10_, and SO_2_ are highly specific predictors of COPD-E severity and disease progression in New Taipei City. (**A**) Predictor screening chart showing the contribution and ranking of each variable in predicting COPD—E severity. Contour profile (upper left), surface profile (upper right), and predictor profiler (lower panel) plots for (**B**) longitude, latitude, (**C**) PM_2.5_, PM_10_, lnc-IL7R, (**D**) SO_2_, and BMI based on the Gaussian process model of %LAA_-950insp_-based COPD-E severity. Fit used Gaussian correlation function. Nugget parameter was set to avoid singular variance matrix.

**Figure 6 biomedicines-09-01833-f006:**
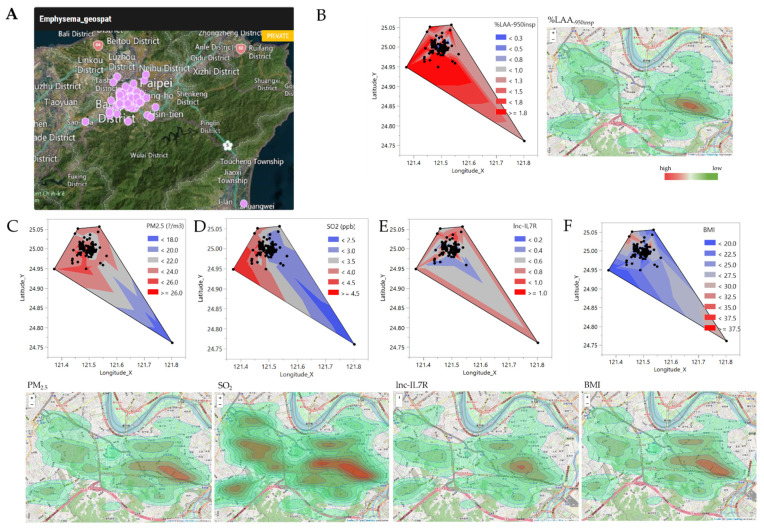
Low BMI, and lnc-IL7R, with concomitant high PM_2.5_, and SO_2_ levels is pathognomonic of exacerbated/severe COPD-E in New Taipei City, Taiwan. (**A**) Image of the geospatial mapping patients with COPD-E in New Taipei City. Contour-based polygon plots and heatmap aerial view maps of case distribution according to (**B**) disease severity, and levels of (**C**) PM_2.5_, (**D**) SO_2_, (**E**) lnc-IL7R, and (**F**) BMI in patients with COPD-E in New Taipei City.

**Table 1 biomedicines-09-01833-t001:** Baseline characteristics of our cohort of healthy and COPD participants (*n* = 168).

Variables	Healthy Controls (*n* = 43)		Patients with COPD (GOLD Stage, *n* = 125)			
	Non-Smoker (*n* = 21)	Smoker (*n* = 22)	I (*n* = 18)	II (*n* = 58)	III (*n* = 38)	IV (*n* = 11)
Age (years)						
Mean ± SD (Min-Max)	68.33 ± 7.02 (50.00–80.00)	67.45 ± 6.75 (47.00–80.00)	69.39 ± 5.78 (61.00–80.00)	67.97 ± 8.91 (41.00–9.00)	71.24 ± 6.95 (56.00–80.00)	67.09 ± 5.38 (61.00–79.00)
Median (IQR)	69.00 (67.00–73.00)	69.00 (65.25–71.00)	68.00 (65.25–71.50)	68.50 (62.25–73.00)	70.50 (67.00–77.25)	66.00 (63.00–69.00)
Sex, *n* (%)						
Male	8 (38.10)	17 (77.27)	17 (94.44)	55 (94.83)	31 (81.58)	9 (81.82)
Female	13 (61.90)	5 (22.73)	1 (5.56)	3 (5.17)	7 (18.42)	2 (18.18)
BMI, kg∙m^−2^						
Mean ± SD (Min-Max)	22.79 ± 2.15 (20.50–28.80)	23.14 ± 2.58 (19.11–29.20)	24.05 ± 3.12 (19.10–29.36)	24.33 ± 4.41 (16.40–34.80)	22.50 ± 3.77 (15.80–36.20)	21.31 ± 3.60 (16.20–27.70)
Median (IQR)	22.00 (21.20–24.00)	22.76 (21.85–23.95)	23.90 (21.63–26.29)	24.14 (21.16–26.60)	22.30 (20.00–24.50)	20.60 (19.90–22.98)
Tobacco Smoking, *n* (%)						
Current smoker	0 (0.00)	13 (59.09)	5 (27.78)	31 (53.44)	11 (28.95)	2 (18.18)
Ex-smoker	0 (0.00)	9 (40.91)	13 (72.22)	23 (39.66)	22 (57.89)	8 (72.73)
Never-smoker	100 (100)	0 (0.00)	0 (0.00)	4 (6.90)	5 (13.16)	1 (9.09)
Smoking pack-years						
Mean ± SD (Min-Max)	0 (0.00–0.00)	65.00 ± 31.43 (30.00–145.00)	48.89 ± 35.19 (5.00–150.00)	49.02 ± 36.34 (0.00–180.00)	49.30 ± 35.66 (0.00–156.00)	56.73 ± 37.65 (0.00–123.00)
Median (IQR)	0 (0.00–0.00)	57.00 (40.00–79.50)	42.50 (20.50–60.00)	40.00 (23.00–60.00)	40.00 (25.00–75.00)	46.00 (35.00–85.00)
Pulmonary function indices						
FEV1 (L) Mean ± SD (Min-Max)	1.98 ± 0.37 (1.26–2.76)	2.18 ± 0.42 (1.74–3.47)	1.95 ± 0.26 (1.55–2.56)	1.67 ± 0.40 (1.01–3.07)	0.98 ± 0.25 (0.61–1.51)	0.61 ± 0.13 (0.43–0.87)
Median (IQR)	1.97 (1.80–2.06)	2.09 (1.90–2.29)	1.90 (1.74–2.11)	1.61 ^b’^ (1.38–1.90	0.99 ^a’b’c’d’^ (0.74–1.12)	0.58 ^a’b’c’d’^ (0.52–0.66)
FEV1 % Mean ± SD (Min-Max)	101.43 ± 5.10 (95.00–117.00)	98.60 ± 6.68 (90.00–111.00)	85.42 ± 5.24 (80.00–97.70)	63.81 ± 8.57 (50.00–79.00)	40.01 ± 5.71 (32.00–49.80)	24.85 ± 3.98 (17.50–29.90)
Median (IQR)	101.00 (98.00–103.00)	96.85 (93.00–103.00)	84.55 (81.3–86.68)	65.00 ^a’b’c^ (57.38–72.00)	39.05 ^a’b’c’d’^ (35.00–45.00)	25.00 ^a’b’c’d’^ (22.10–27.95)
FEV1/FVC % Mean ± SD (Min-Max)	100.76 ± 9.42 (80.00–125.00)	100.45 ± 8.01 (90.00–120.00)	63.49 ± 4.05 (54.64–68.26)	59.33 ± 6.86 (45.00–69.72)	47.93 ± 8.60 (28.00–65.00)	38.84 ± 8.41 (27.00–49.61)
Median (IQR)	100.00 (98.00–105.00)	98.00 (95.25–107.75)	63.68 (61.25–66.87)	59.25 ^a’b’^ (54.12–65.50)	46.50 ^a’b’c’d’^ (42.11–55.25)	41.41 ^a’b’c’d’^ (30.93–45.67)
Emphysema severity						
Null/Mild (%)			66.67	19.05	0.00	0.00
Moderate (%)			33.33	66.67	69.23	20.00
Severe (%)			0.00	14.28	30.77	80.00

COPD, chronic obstructive pulmonary disease; GOLD, Global Initiative for Chronic Obstructive Lung Disease; M, male; F, female; FEV1, forced expiratory volume in 1 s; FVC, forced vital capacity; BMI, body mass index; IQR, interquartile range. The values of FEV1/FVC % and FEV1 % were analyzed by Kruskal–Wallis tests and Dunn’s multiple comparisons (^a’^ *p* < 0.01, compared with non-smoker; ^b’^ *p* < 0.01, compared with smoker; ^c^ *p* < 0.05, ^c’^ *p* < 0.01, compared with COPD patients with GOLD stage I; ^d’^ *p* < 0.01, compared with COPD patients with GOLD stage II.

**Table 2 biomedicines-09-01833-t002:** Gaussian process model of %LAA-950insp-based COPD-E severity and progression: *lnc-IL7R*, PM_2.5_, and PM_10_.

Column	Theta	Total Sensitivity	Main Effect	lnc-IL7R Interaction	PM10 Interaction	PM2.5 Interaction
lnc-IL7R	0.0003	0.1667	0.1667	-	0	5.35 × 10^−9^
PM10	4.82 × 10^−9^	0	0	0	-	0
PM2.5	1.70 × 10^−9^	0.8333	0.8333	5.35 × 10^−9^	0	-
**μ**	**σ^2^**	**Nugget**				
1.3155	470.0283	0.001				
**−2*Loglikelihood**						
240.6087						

**Table 3 biomedicines-09-01833-t003:** Gaussian process model of %LAA-950insp-based COPD-E severity and progression: longitude, and latitude.

Column	Theta	Total Sensitivity	Main Effect	Longitude Interaction	Latitude Interaction
Longitude	0.0046	0.9986	0.9986	-	0
Latitude	0.0003	0.0014	0.0014	0	-
**μ**	**σ^2^**	**Nugget**			
1.4838	520.33	0.001			
**−2*Loglikelihood**					
285.8087					

**Table 4 biomedicines-09-01833-t004:** Gaussian process model of %LAA-950insp-based COPD-E severity and progression: BMI, and SO_2_.

Column	Theta	Total Sensitivity	Main Effect	BMI Interaction	SO_2_ Interaction
BMI	8.21 × 10^−6^	0.6833	0.6833	-	2.74 × 10^−7^
SO2	8.75 × 10^−5^	0.3167	0.3167	2.74 × 10^−7^	-
**μ**	**σ^2^**	**Nugget**			
1.1118	390.41	0.001			
**−2*Loglikelihood**					
248.4064					

## Data Availability

The data used in the current study are all contained in the manuscript, and may be obtained upon reasonable request from the corresponding author.

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
