# Peer review of "Determinants of Pulmonary Emphysema Severity in Taiwanese Patients with Chronic Obstructive Pulmonary Disease: An Integrated Epigenomic and Air Pollutant Analysis"

_biomedicines, 2021, doi:10.3390/biomedicines9121833_

Round 1
Reviewer 1 Report
Sheng-Ming Wu and colleagues demonstrated that lower BMI and lnc-IL7R, with concomitant higher PM2.5, PM10, and SO2 levels are pathognomonic of exacerbated/aggravated COPD in Taiwan.
Very interesting data with a lot of statistical analysis, and especially geospartial mapping presentation is impressive. These findings potentially help environmental health policy formulation for lowering disease risk and severity in Taiwan as authors indicated. I am not sure this concept can be applied to other countries or not (need to discuss).
However, I have several concerns on this paper.
Firstly, Table 1 looks really messy, and difficult to understand, and authors can find the better way to present. For example, put median underneath mean+/-SEM. Also authors need to add additional Table regarding to the relationship between GOLD guideline and prevalence of emphysema. It was stated in L218-227, but it is difficult to catch. Also no explanation on Y axis in Figure 1, what is 0, 1, 2. I understand authors categorised as described in method section, such as no or mild emphysema (%LAA-950insp < 6), moderate emphysema (6 ≤ %LAA-950insp < 14), and severe emphysema (%LAA-950insp ≥ 14), but no explanation on 0-2 numbering. Please include those in Figure legend..
Secondary, authors mixed up factors affecting the outcome. Lnc-IL7R (and possibly BMI) are biomarkers, but PM2.5, PM10, SPO2 are estimated exposure (not biomarkers). Authors need to discuss carefully in Discussion section. For bridging, blood oxidative stress marker is useful to interpret the relationship between air pollution and biomarkers. Hope authors include the data if available, or consider this in future. Discuss as limitation of study
Thirdly, authors need to use the words of disease name carefully. I was confused that authors used the words of “COPD” and “COPD-E” at random sometimes. In conclusion section (even in abstract), it should be COPD-E rather than COPD (not necessary to be generalised).
Lastly, To develop COPD or emphysema, previous exposure of PM2.5 and other gases (20-30 years ago) will be more important than current on-going exposure. Please discuss.
Minor
Have you captured CT image from healthy subjects, didn’t you find any senile emphysema in elderly? Or emphysema without any decline of lung function?
Reviewer 2 Report
I read the present manuscript with great interest and i thnk that this is an excellent, well-written article, with robust data.
I have a few comments:
- Please adjust the article format. Some characters are bigger than others
- Please make a deep english revision, i found some minor errors throughout the text
- If Ambient air pollutant exposure data are publicly available, please cite the source.
- Please specify whether sample size evaluation was estimated.
Round 2
Reviewer 1 Report
Authors have addressed most of my comments/suggestions appropriately. Especially Table 1 has been improved impressively.